# Application of Head Loss Coefficient for Surcharge Straight Path Manhole to Improve the Accuracy of Urban Inundation Analysis

Changjae Kwak [1] , Jungsoo Kim [2,*], Sungho Lee [3] and Ingi Yoo [3]

[1]   National Disaster Management Institute, Ulsan 44538, Korea
[2]   Department of Civil Engineering, Bucheon University, Bucheon 14623, Korea
[3]   Research Institute, C&I Tech, Incheon 21556, Korea
*   Correspondence: hydroguy@bc.ac.kr; Tel.: +82-32-610-3315

**Abstract:** Currently, adopted runoff analysis models focus on the characteristic factors of watersheds and neglect the analysis of the flow in conduits. Additionally, the usually employed XP-SWMM modeling package generally underestimates the flood area because it considers manholes as nodes and does not consider local head losses according to the shape and size of the nodes. Therefore, it is a necessity to consider the loss coefficient in surcharge manholes to improve inundation and runoff analysis methods. This study aims at improving the accuracy of discharge analysis before analyzing the storage and runoff reduction effects of storage facilities. Hydraulic experiments were conducted according to the changes in discharge and manhole shapes. We show that the flood area increases as the overflow discharge at manhole increases due to the application of the head loss coefficient. We demonstrate a concordance rate ≥95% between results and observed flood area when accurate input data (from the parameters of the target watershed) and the head loss coefficient (from hydraulic experiments) are applied. Therefore, we demonstrate that the result of our 2D inundation analysis, considering the head loss coefficient in surcharge manhole, can be used as basic data for accurately identifying urban flood risk areas.

**Keywords:** drainage system; flood area; head loss coefficient; surcharge manhole



## 1. Introduction

Urban watersheds that have been highly developed through rapid urbanization and industrialization are shortening the arrival time of urban flood volume and increasing the peak discharge by increasing impervious areas and decreasing the surface roughness [1]. Moreover, torrential rains are repeatedly occurring at frequencies exceeding the design frequency of existing sewer facilities, which is rapidly lowering the retention and exclusion capacity of installed sewer systems [2]. It follows that the damage inferred by inland flood inundation is increasingly becoming more important than that caused by external flooding [3]. Several measures to reduce urban flooding are being established, such as increasing the design frequency of sewer systems, installing rainwater runoff reduction facilities, and planning large-scale rainwater storage facilities. However, most of these measures remain insufficient, as it is not possible to accurately determine the discharge capacity of installed sewers [4]. Moreover, installing runoff reduction facilities at the right place is challenging, especially since all the existing sewer systems have been planned and designed in open channel condition [5]. Hence, the flow of surcharged sewer systems and the consideration of surcharged sewer runoff have not been studied enough in the literature [6]. Consequently, inland flood inundation countermeasures based on a more accurate urban flooding analysis and a relatively accurate simulation of urban runoff in consideration of the surcharge condition are required for the accurate evaluation of drainage capacity of sewer systems and proper design and construction of storage facilities. Among the many models widely used for urban runoff and inundation analysis, the XP-SWMM model is frequently adopted because it can perform surface runoff, sewer flow,

and 2D inundation analysis. The XP-SWMM model, based on the EPA SWMM engine, is an analysis-programming package capable of performing sewer and storm-water analysis of urban watersheds and inundation networks simulation. It can also perform 1D and 2D analyses simultaneously. However, the XP-SWMM model is limited in the consideration of manholes as nodes, without any capability to consider flow changes and energy losses induced by changes in the shape or size of manholes. Hence, it is recommended to input the head loss coefficient as a node in the analysis of sewer runoff due to the slope of the sewers [7]. Accordingly, when inundation simulation of XP-SWMM model is performed, to design the urban watershed drainage system and evaluate sewer capacity, the model is tested using the roughness coefficient of the pipeline without considering the head loss coefficient, mainly because the manhole's head loss coefficient cannot be determined despite the difference in the flood areas between the case in which the head loss coefficient is applied to the surcharge manhole and the case where it is not applied. Thus, existing urban runoff sewer analysis models, including the XP-SWMM model widely used in Korea, cannot consider local head losses due to the installation of manholes. This is because they do not reflect the size and shape of manholes and that research on the application of head loss coefficient is still insufficient [8]. Therefore, for a more realistic estimation of the flood area through the XP-SWMM model, two improvements must be advanced. First, a detailed study of inland flood inundation analysis for urban watersheds that can consider energy loss from surcharged manholes should be conducted. Second, standards for inland flood inundation analysis considering sewer systems such as manholes must be established for the calculation of the overflow discharge from sewer analysis.

Among the studies related to inundation analysis in urban areas using the SWMM model, Hsu et al. (2000) simulated inundation caused by surplus water at the nodes of the network using the SWMM model [9]. Choi et al. (2004) considered the reservoir water level and backflow phenomena to improve inundation around the Incheon Bridge reclaimed land using XP-SWMM [10]. Beak et al. (2005) compared the hydraulic behavior characteristics using mathematical model experiments and numerical simulations in order to verify the accuracy of the analysis when the flow of the irrigation conduit is subject to the full water of the sewer [11]. Phillips et al. (2005) proved the applicability of 2D analysis for urban inundation through the connection of the XP-SWMM model and the TUFLOW engine for 2D inundation analysis of urban drainage systems [12]. Lee et al. (2006) calculated the hourly inundation depth and flood area regarding overflow discharge that occurs when rainfall exceeding the capacity of the drainage system occurs [13]. They also considered the case where drainage is poor due to an increase in the external water level by developing and applying a linkage model between the SWMM model and a DEM-based inundation analysis model. Smith et al. (2006) simulated flood areas in urban watersheds by dividing them into sub-watersheds and determined the drainage area for each district [14]. Lee and Yeon (2008) simulated the inundation depth and flood area of urban areas using the XP-SWMM model and analyzed flood area according to the effects of buildings by major time periods [15]. Rangarajan et al. (2008) proved the effect of 2D flood analysis by combining the XP-SWMM and the TUFLOW engine to study the target watershed in Virginia, USA [16]. Kim et al. (2010) investigated the inundation reduction effect of the installation of underground storage tanks by conducting 2D inundation analysis based on discharge analysis and the calculation of overflow discharge using the SWMM model [17]. Ahn et al. (2014) devised a watershed superposition method that adds layers of a 1D storm-water pipe network and considered the inflow efficiency of road surface flow according to the slope [18]. The authors performed inundation simulation considering a branch-type urban watershed (Sadangcheon watershed) by applying their method. Rosa et al. (2015) derived similar results to observed values for flow rate, which was underestimated by the SWMM in the LID watershed, by adjusting parameters such as the saturated hydraulic conductivity, Manning's roughness coefficient, and the initial soil moisture of wetland [19]. Del Giudice and Padulano (2016) selected parameters that have significant effects on the runoff results, such as runoff coefficient and storage constant through sensitivity analy-

sis [20]. Furthermore, Kim et al. (2019) analyzed the inundation pattern according to the DEM (Digital Elevation Models) grid sizes of $1 \times 1$ m and $10 \times 10$ m during 2D inundation analysis using the XP-SWMM model [21]. They showed that the precision on the determination of the inundation depth improves by considering smaller grids size. Hasan et al. (2019) conducted an analysis for predicting flash floods, which are increasing in frequency in tropical regions [22]. They also analyzed efficient inundation reduction plans using the XP-SWMM model. Shen and Tan (2020) analyzed the applicability of resampled DEM through inundation simulation with a resolution improved by resampling technology [23]. The authors showed that VSL shortened the simulation time in a 1D/2D discharge analysis model that combined the SWMM and P-DWave 2D models to improve the errors of the overestimated flood area due to buildings in the inundation analysis for urban areas. Choo et al. (2021) analyzed the flood-reduction effect when dams, sluice gates, and culverts are installed in rivers around an urban area using the SWMM model [24]. Furthermore, Parnas et al. (2021) performed sensitivity analysis for Green-Ampt, Horton, and Holtan methods, which are inundation analysis methods for pervious areas to increase the accuracy of urban discharge analysis using the SWMM model and the STORM model [25]. The authors showed that the Holtan method exhibited the best results in discharge behavior from long-term simulations.

As shown in the above examples, many national and overseas studies are attempting to analyze inundation simulations more practically in urban watersheds. However, existing analytical studies on the application of energy loss due to the surcharge of the sewer system in urban watershed discharge and inundation analysis are lacking. Similarly, studies on the effect of energy loss according to the size and shape of manhole in the surcharge sewer flow on water flow capacity are insufficient. Moreover, no study on the application of head loss coefficient in surcharge sewer system (manhole) in inundation analysis has been reported so far.

Therefore, this study calculated the head loss coefficient in surcharged manholes using hydraulic experiments and directly applied the loss coefficient to inundation simulation for urban watersheds. For application of the head loss coefficient in surcharged sewers estimated through discharge, analyses of the XP-SWMM were first performed, and then, re-discharge analysis was conducted by applying the calculated head loss coefficient to the estimated surcharge sewer. To analyze the accuracy of the flood area, 2D inundation analysis was performed using the result of the re-discharge analysis. The accuracy of the spatial location was analyzed by applying the simulated flood area of the target watersheds and the Lee–Sallee shape index (LSSI) to the inundation trace map provided on the public data portal of the Ministry of the Interior and Safety. Lastly, the applicability of the head loss coefficient to the urban runoff model has been demonstrated to more precisely calculating the flood area in the 2D inundation analysis of the XP-SWMM model.

## 2. Calculation of Head Loss Coefficient at Manhole in Surcharge Straight Path

### 2.1. Loss of Surcharge Manhole

In general, the surcharge in a manhole corresponds to the case of a continuous full sewer flow in full connecting pipe from the well height to the manhole to the upper height of the manhole that does not overflow. In this study, the start point of surcharge condition is attributed to the moment when the inflow and outflow pipes become full, and the manhole's water depth exceeds the diameter of the connecting pipe. Head loss is caused by the fluctuation of the water surface due to the effect of vortex flow inside the manhole in surcharge condition [26]. Sangster et al. (1958) proposed Equation (1) for calculating the head loss coefficient from the continuity and momentum equations of the flow at the manhole inlet and outlet [27]. Equation (1) was later used to calculate the head loss coefficient inside a manhole by Marsalek (1984), Bo Pederson and Mark (1990), and Shinji and Tetsuya (1998), among others [28–30].

$$\Delta h = K \frac{V^2}{2g} \tag{1}$$

here, $\Delta h$ (m) is the head loss difference between the manhole inlet and outlet, $K$ (m/m) is the head loss coefficient inside the manhole, and $V$ (m/s) is the mean flow velocity of the sewer (Figure 1).

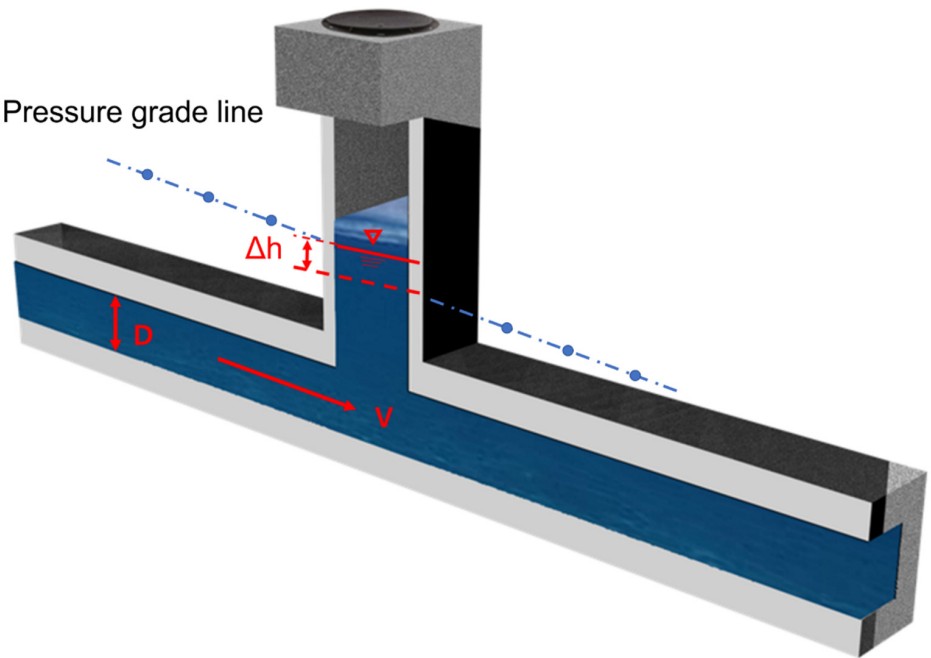

**Figure 1.** Head Loss at Manhole.

### 2.2. Hydraulic Experiments

To apply the head loss coefficient in the surcharged manhole to the XP-SWMM model, hydraulic experiments are required. In this study, hydraulic model experiments were conducted for circular and square manholes, which are mainly used in storm-water drainpipes to calculate the head loss coefficient of surcharged manhole. The specifications of the standard (circular) manhole no. 1 (900 mm in diameter) and the special (square) manhole no. 1 (900 × 900 mm) suggested in the Sewage Facility Standard (Ministry of Environment, 2019) were selected, and 1/5 down-scaled models of the manholes were fabricated (Figures 2 and 3) [31].

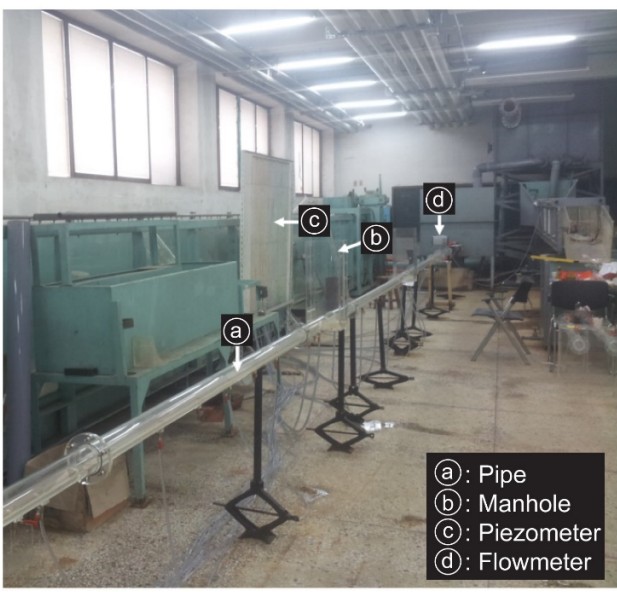

**Figure 2.** Experimental apparatus for the estimation of head loss coefficients.

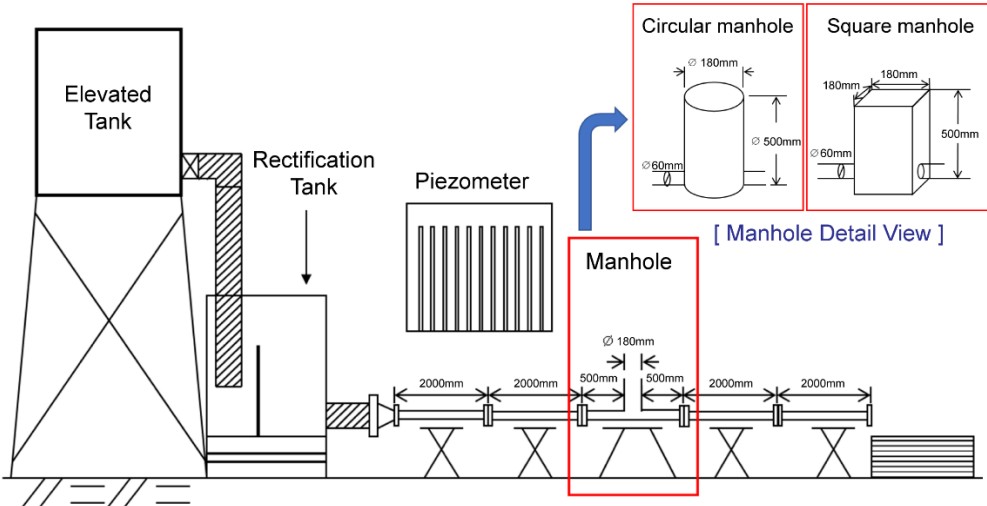

**Figure 3.** Installation diagram of the hydraulic experiment model.

As shown in Figures 2 and 3, the dimensions of inflow/outflow pipe were set to 450 cm in length and 60 mm in diameter. The pressure changes inside the sewers were measured by installing manometers at 30 cm intervals along the pipes joining the manholes. In particular, the separating intervals distance of manometers was reduced to 10 cm before and after the inflow and outflow pipes at the connection point with the manhole for more accurate local measurement of pressure changes. The piezometers were a clear acrylic pipe with a 5 mm internal diameter and connected to a manometer board, which allowed the reading of the piezometric heads with an accuracy of ±1 mm. The number of experiments that were carried out under steady-state conditions were 19 cases. All processes were repeated until every experimental case was finished along with the already-mentioned steps. In order to reduce the measurement error of the pressure head, the pressure head was measured at least three times for each experimental condition, and the measured values were averaged. A collection container with a length of 90 cm, a width of 80 cm, and a height of 70 cm was installed at the lower end of the outflow pipe to measure the experimental flow rate. The experimental flow rate was varied from 0.001 to 0.003 m$^3$/s to measure the change of the head loss coefficient according to the flow rate change and to determine the mean head loss coefficient [8]. As a calculation criteria for the experimental flow rate, the ideal flow rate (1.0 m/s) in the storm-water sewer suggested in the Sewage Facility Standard (Ministry of Environment, 2019) was selected as the minimum flow rate [31]. In addition, the flow rate at which the manhole does not overflow was selected as the maximum flow rate. The minimum and maximum flow rates of the 1/5 down-scaled hydraulic model were calculated by applying the Froude's Law of Similarity to the selected flow rates.

### 2.3. Calculation of the Head Loss Coefficient

The depth of the manhole is varied as an experimental condition. The head loss coefficient at the manhole was calculated by applying the pipe average flow velocity calculated from the head loss (Δh) and inflow rate in the surcharge manhole in Equation (1). The calculated coefficients, shown in Figure 4, are dependent on the change in the ratio of the manhole water depth (h) and inflow pipe diameter (D). Experimental head losses were plotted the observed head loss (Δh) and velocity head for individual manholes. A sample of those graphs is given in Figure 5. The head loss coefficients were then taken as the slope of the regression lines fitted to the experimental data. These coefficients and their 95% confidence limits are given in Table 1.

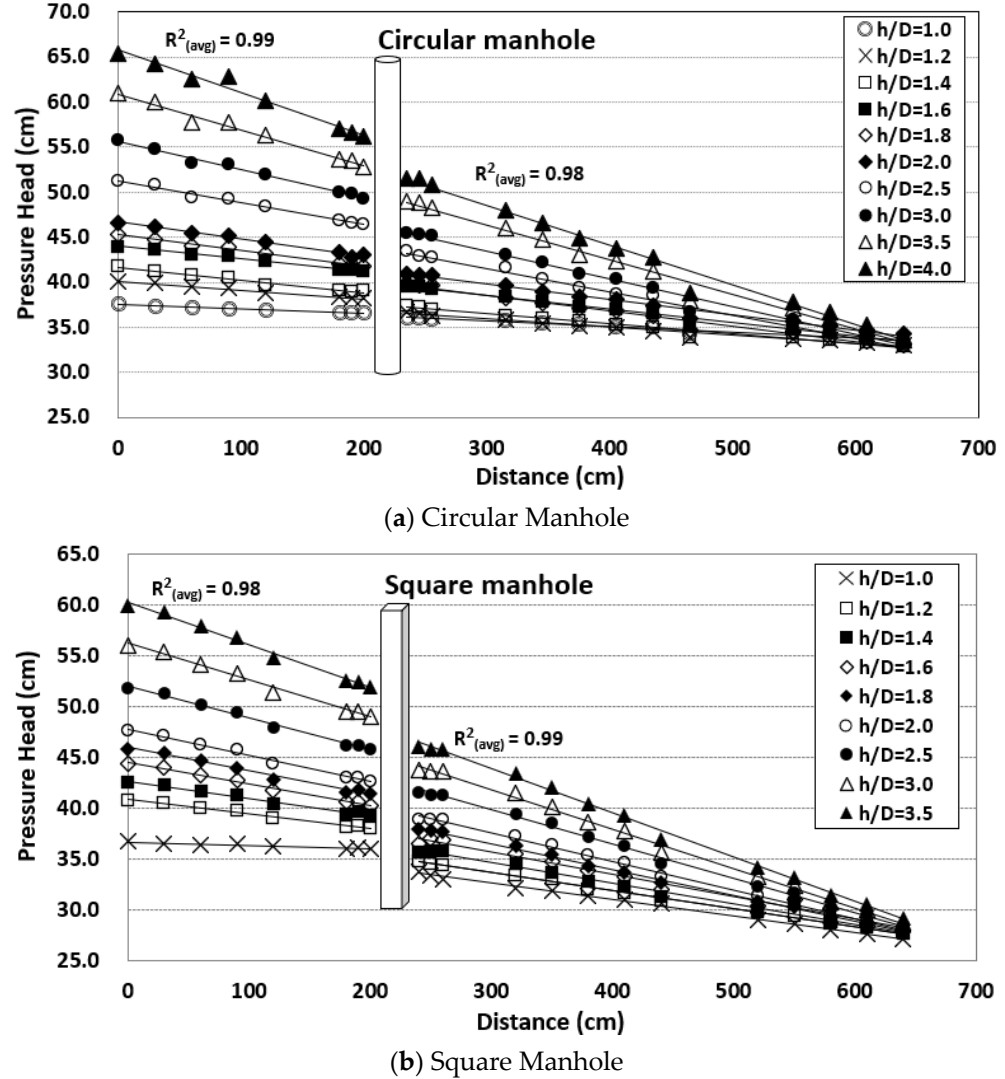

**Figure 4.** Change of head loss at surcharge manhole with variation of inlet discharge.

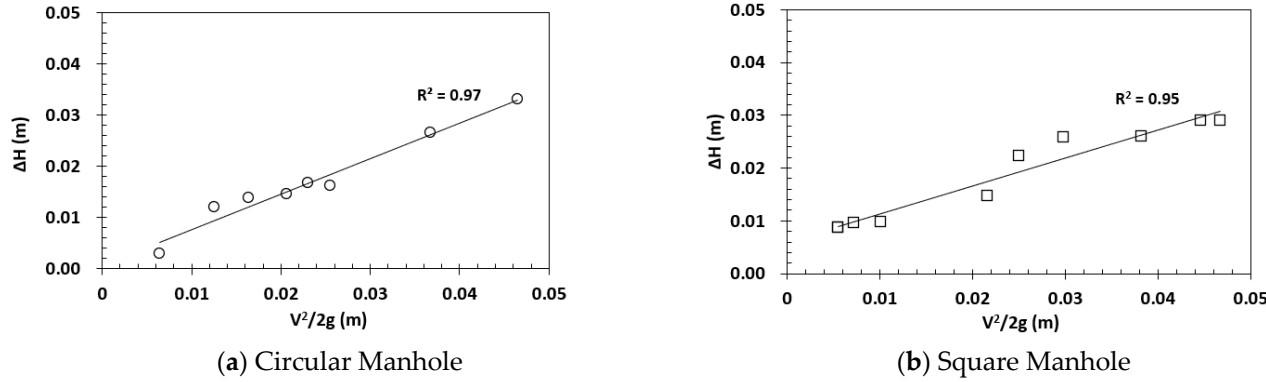

**Figure 5.** Relationship between Head Loss ($\Delta h$) and Velocity Head ($V^2/2g$).

**Table 1.** Head loss coefficient for surcharge manhole.

| Manhole Type | K |
|---|---|
| Circular-Mean values 95% confidence limits | 0.61 (0.529~0.697) |
| Square-Mean values 95% confidence limits | 0.68 (0.587~0.773) |

*2.4. Application of Head Loss Coefficient in Inland Flood Inundation Analysis*

In general, inundation simulation is performed without considering loss for surcharged sewer systems in the 2D inundation analysis of XP-SWMM, which is used in inland flood inundation analysis. Nonetheless, when the sewer performance and inundation effect in urban areas are evaluated using the XP-SWMM model, if the head loss coefficient is applied to surcharge manhole, additional surcharge occurs because it affects not only the surcharge manhole but also the surrounding manholes. The head loss increases among some of the existing surcharge manholes, and the occurrence of overflow is simulated. This implies that the evaluation results may differ depending on whether or not the head loss coefficient is applied (using the XP-SWMM) in surcharge manhole when evaluating the drain capacity of the existing sewers and the occurrence of inundation in urban watersheds. Therefore, novel inland flood inundation analysis methods considering energy loss in surcharge manhole are required for the Comprehensive Plan for Storm and Flood Damage Reduction and storm-water storage tank installation project. The project selects domestic disaster risk districts and presents preventive measures through inland flood inundation analysis by frequency.

In general, the XP-SWMM sets up a pipe network by considering each manhole as a node and cannot reflect losses generated in the manhole structure itself. Hence, this study selected a method of replacing the head loss coefficient of manhole by applying the inlet and outlet head loss coefficient of surcharge sewer before and after a node.

Furthermore, the same simulation results were obtained when the head loss coefficient was applied to the inlet loss, to the outlet loss, and equally to inlet and outlet losses. Therefore, we judged it more rational to consider the inlet and outlet losses separately. The inundation analysis process considering the head loss coefficient is shown in Figure 6. The flood area simulation process applying the head loss coefficient is generally the same as the 2D inundation simulation method of the XP-SWMM, which is used in practice. However, after the first discharge analysis, an additional 2D flood area analysis process is conducted by applying the head loss coefficient to the surcharge sewer estimated by checking the its cross-section in the simulated result.

The head loss coefficients, which can be considered in the SWMM model (including XP-SWMM), include scale down/up head loss, energy head loss, and pressure change coefficients, which can be considered as inlet/outlet head loss coefficient, among other coefficients [32]. The scale down/up head loss coefficients model the loss of velocity when the shape of the cross-section changes suddenly until the next turbulence in one sewer. They are only used in the sewers of the hydraulics layer. For inlet/outlet head loss coefficients, the energy head loss and the pressure change coefficient can be considered. First of all, the energy head loss coefficient is composed of inlet loss and outlet loss, and the inlet and outlet head loss coefficient is the value of the square of the velocity applied to the sewer's inlet and outlet ($K\frac{V^2}{2g}$).

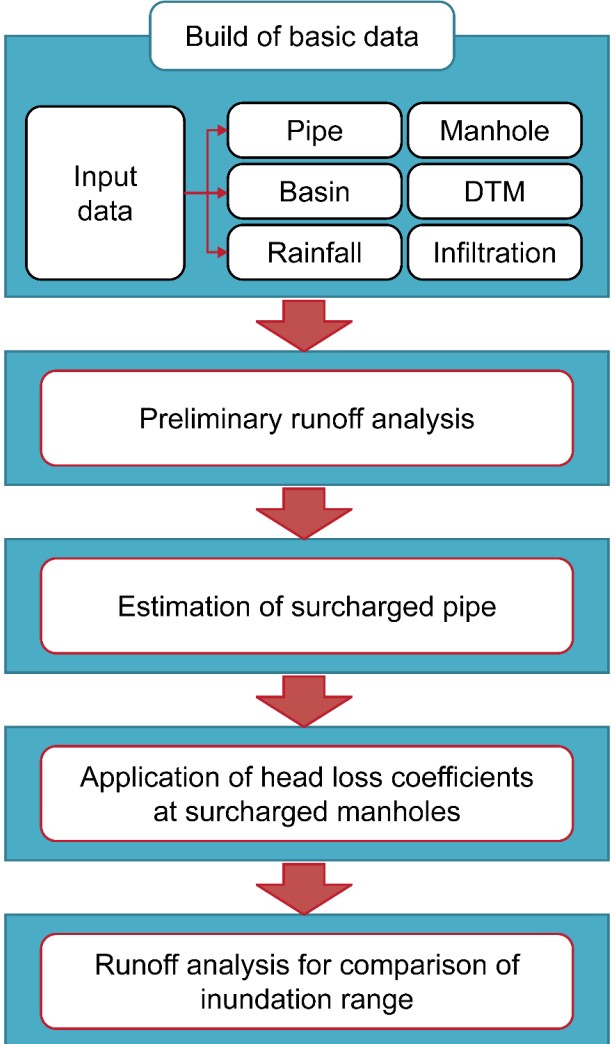

**Figure 6.** Flow analysis process in inundation area.

Because only the continuity equation is used for the junction, losses are practically modeled by the momentum equation for the sewer. The energy head loss coefficient is also used only for the sewers and hydraulics layer. The pressure change coefficient ($K_u$) is converted and modeled as energy loss using the following momentum equation:

$$B = \frac{V_u}{V_0} \tag{2}$$

$$K' = K_u - 1 + B^2, \tag{3}$$

where $V_u$ and $V_o$ denote the upstream and downstream comparison velocities, respectively. $K'$ denotes the equivalent energy loss. Lastly, the other head loss coefficient is additionally considered when the square of velocity ($K\frac{V^2}{2g}$) is applied only for the sewers and hydraulics layer [7].

## 3. Result and Discussions

### 3.1. Analysis of the Effect of the Head Loss Coefficient in Flood Area Analysis

In this study, Dorim 1, Gunja, and Gildong Watersheds located in Seoul were selected as target watersheds to conduct inland flood inundation analysis by applying the head loss coefficient of surcharge manhole (Figure 7). The effect of head loss coefficient when conducting flood area analysis is investigated by applying the head loss coefficient after calibrating

the parameters of the target watersheds. Dorim 1, Gunja, and Gildong watersheds have a history of many flood damages due to torrential rains. Thus, their inundation trace maps are prepared and available, and they can be used as basic data for flood area comparison. Furthermore, they are areas where pump station inflow and sewer flow measurement data are being collected. This supported their choice for this study, as accurate research data could be derived because it was easy to estimate the parameters of the watersheds using pump station inflow data and sewer flow data. The applied rainfall for inundation analysis was of 259 mm for two days, and the rainfall data on 20 and 21 September 2010 were used for this investigation (Figure 8).

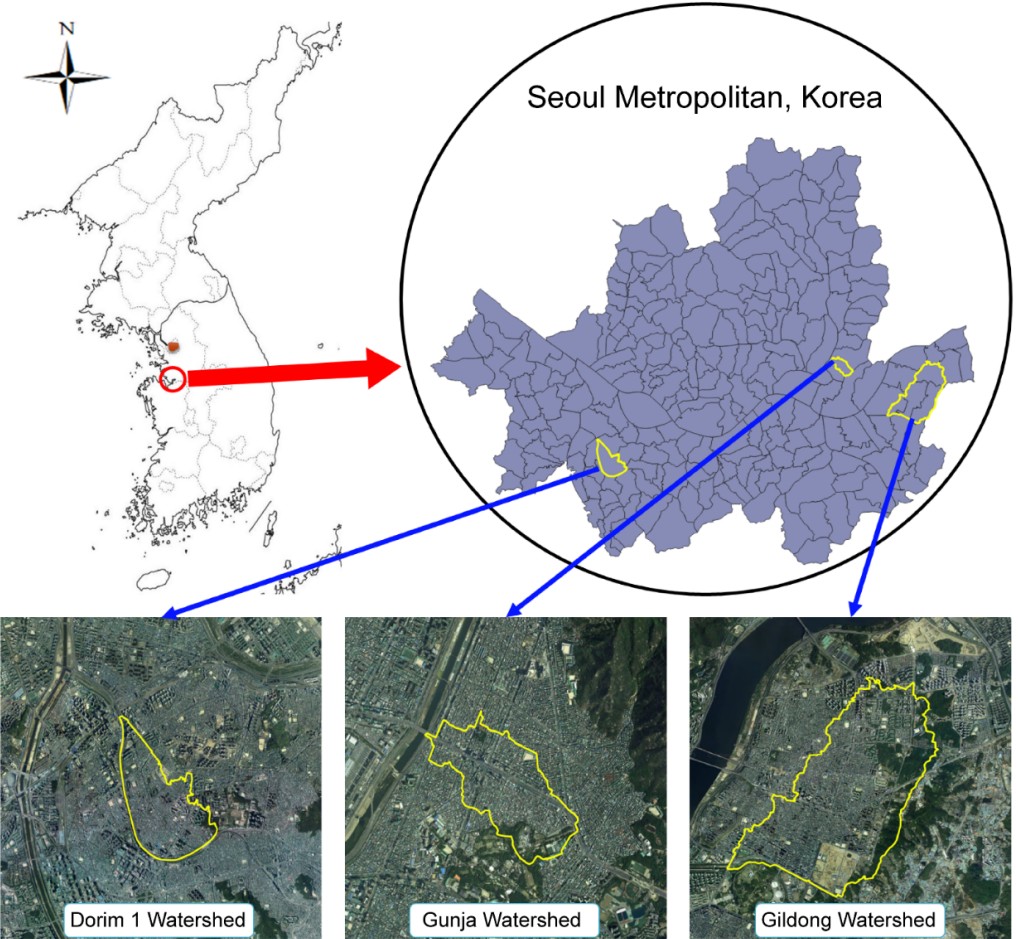

**Figure 7.** Target watersheds for inundation analysis.

### 3.2. Current Status of Target Watersheds and Correction of Parameters

3.2.1. Dorim 1 Watershed

The Dorim 1 Watershed is located in the downstream of the Dorimcheon and spans Yeongdeungpo-gu and Dongjak-gu, Seoul. Its watershed area is 2.7063 km$^2$, and there is a history of an area of 0.3565 km$^2$ being flooded due to a rainfall on 20 and 21 September 2010 (Figure 9a).

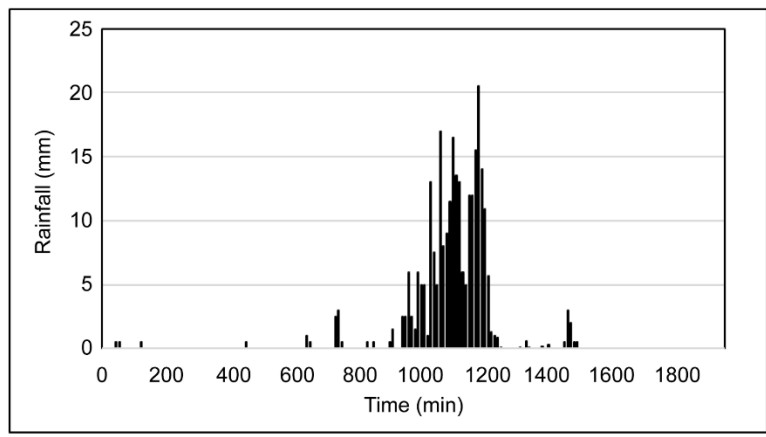

**Figure 8.** Hyetograph at Seoul AWS (20~21 September 2010).

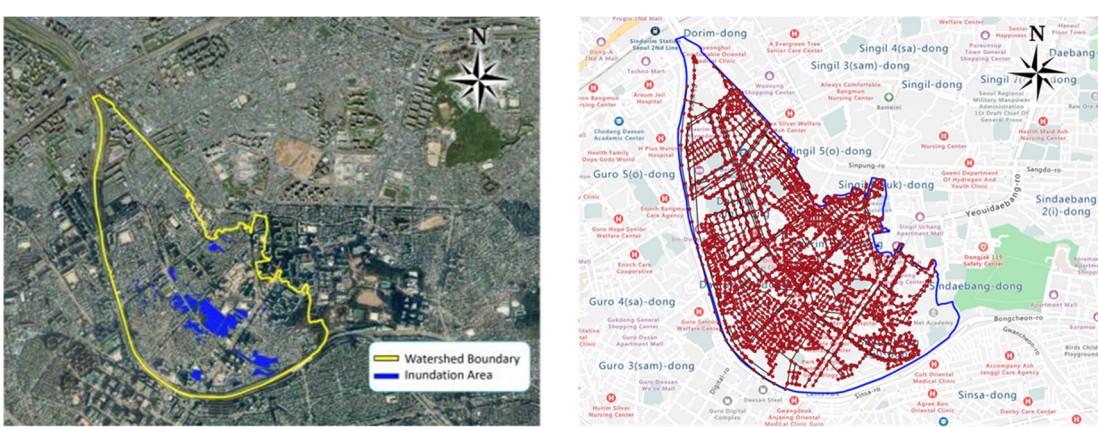

(a) Inundation Area (20-21 September 2010)    (b) Sewer Pipe network Input Data

**Figure 9.** GIS map of the Dorim 1 watershed.

To set up a 1D pipe network in the XP-SWMM model, the input data were composed using the pipe network data and a numerical map. For this study, the input data were constructed considering the operating conditions of Daerim 2 and Daerim 3 rainwater pump stations, which consisted of 3112 nodes and 3364 links (Figure 9b). To conduct a 2D inundation analysis using the XP-SWMM model, topographic data were constructed by generating a digital terrain model (DTM) using numerical maps on a scale of 1:5000. A triangular irregular network (TIN) was constructed after generating a shape file by selecting contour and elevation point layers, which have an elevation property from the numerical map of the target watersheds. Then, the topographic data were constructed by converting it to a DTM in XP-SWMM. The grids were composed of approximately 26,000 cells of a $10 \times 10$ m size each.

The input data of the XP-SWMM model were calibrated using the inflow data of the Daerim 3 rainwater pump station located in the Dorim 1 watershed, during the rainfall events on 20 to 21 September 2010 and on 26 to 29 July 2011.

Figure 10 shows the comparison between the pump inflow hydrologic results and the simulated hydrologic curve of the Dorim 1 watershed, which is a target area for rainfall events on 21 September 2010 and 26 July 2011 according to the correction. The simulated hydrologic curve indicates that the maximum pumping inflow during the rainfall event on 20 to 21 September 2010 was 28.52 m$^3$/s, and the simulated peak discharge was 31.22 m$^3$/s, with a good relative error of about 5.6%. The maximum pumping inflow during the rainfall event on 26 to 29 July 2011 was 32.49 m$^3$/s, and the simulated peak discharge was 27.99 m$^3$/s, with an

error of about 13%. However, the overall shape of the discharge hydrologic curves excluding the peak discharge were similar, indicating relatively accurate values.

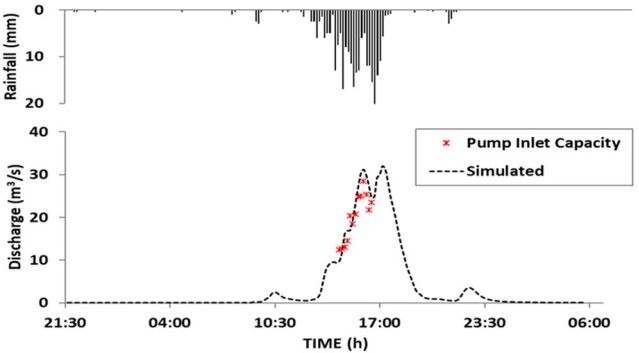

(a) Comparison of Observed Discharge and Simulated Discharge (20-21 September 2010)

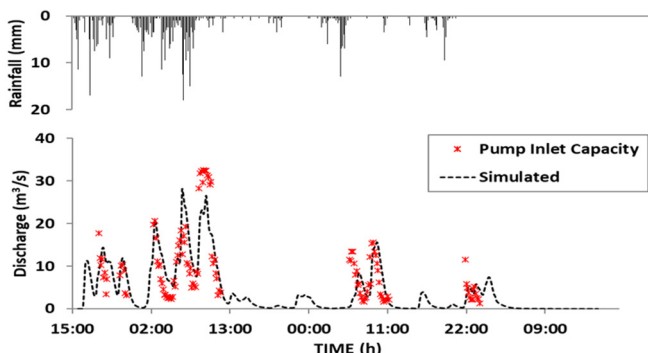

(b) Comparison of Observed Discharge and Simulated Discharge (26-29 July 2011)

**Figure 10.** Input parameters estimation of the Dorim 1 watershed.

Table 2 shows the correction results of the major parameters for the Dorim 1 watershed in the XP-SWMM model. Here, the parameters of the SWMM model are divided into topographic and hydrologic parameters. For the topographic parameters, 1/5000 topographic and GIS maps were used. For the hydrologic parameters, the correction range from the lower to the upper limit values of the parameter was used because of the broad range according to the user's subjective judgment and the same material. In particular, the ultimate infiltration rate was estimated by considering four soil types, and different values were applied depending on the soil type.

**Table 2.** Correction results of the main parameters (Dorim 1 watershed).

| Main Input Parameters of XP-SWMM | | Initial Parameters Value | Correction Parameters Value |
|---|---|---|---|
| Catchment parameters | Width (m) | 2.24~114.02 | 2.24~114.02 |
| | Impervious area (%) | 1.6~100 | 1.6~100 |
| | Slope (m/m) | 0.000~36.980 | 0.000~36.980 |
| Channel parameters | Circular Size (D mm) | D200~D1100 | D200~D1100 |
| | Rectangular Size (m × m) | $1.0 \times 1.0$~$3.0 \times 2.5$ | $1.0 \times 1.0$~$3.0 \times 2.5$ |
| | Manning roughness of channel | 0.014~0.020 | 0.020 |
| Infiltration parameters | Decay rate (Horton's) (1/s) | 0.001 | 2.0 |
| | Ultimate infiltration rate (Horton's) (mm/h) | 10.0 | 2.5~25.4 |
| | Manning roughness of impervious area | 0.014 | 0.014~0.015 |
| | Depression storage of impervious area (mm) | 0.0 | 2.0 |

### 3.2.2. Gunja Watershed

The Gunja watershed is located in the downstream of the left bank of the Jung-nangcheon. The rainwater generated in the watershed flows out to the Han River. The watershed area is 0.9673 km$^2$, and there is a history of an area of 0.1917 km$^2$ around the Gunja Station being flooded by the rainfall event on 20 to 21 September 2010.

The input data were configured using the pipe network data and numerical maps of Seoul Metropolitan City to set up the 1D pipe network of the XP-SWMM model. For this study, a drainage watershed with an area of 96.6 ha was divided into 43 sub-watersheds, which consisted of 44 nodes and 43 links (Figure 11). The topographic data were constructed

by generating a DTM suing a numerical map of 1:5000 scale, the same as the Dorim 1 Watershed. The grids were composed of about 18,000 cells of the size $10 \times 10$ m.

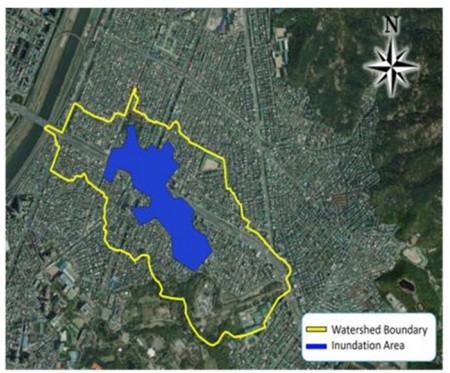
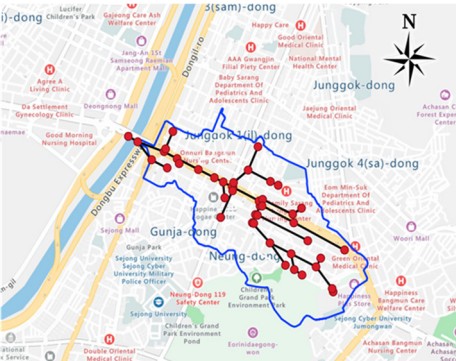

(a) Inundation Area (20-21 September 2010)  (b) Sewer Pipe network Input Data

**Figure 11.** GIS map of the Gunja watershed.

The input data of the XP-SWMM model were corrected using discharge data that were observed from a flowmeter installed at the bottom outlet point in the Gunja Watershed by the Seoul Waterworks Research Institute. The XP-SWMM model was calibrated using the runoff of the rainfall event on 12 June 2010 observed at the bottom outlet point of the pipe network in the Gunja Watershed.

Figure 12 shows the comparison result of observed and simulated hydrologic curves at the bottom outlet point of the Gunja Watershed, which is a target area, for the rainfall event on 12 June 2010 after correction. The observed peak discharge was 2.077 m$^3$/s, and the simulated peak discharge was 1.997 m$^3$/s. Thus, the relative error of the observed and simulated peak discharges was about 4%, showing a good error. The simulated values matched relatively well with the observed values in the discharge hydrologic curve for corrected rainfall events.

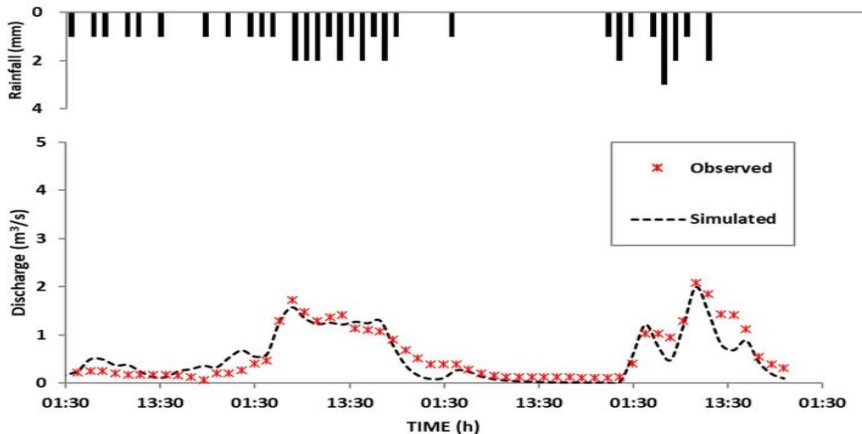

**Figure 12.** Comparison of Observed Discharge and Simulated Discharge (12 June 2010).

Table 3 shows the correction result of major parameters of the XP-SWMM model using the corrected rainfall event on 12 June 2010. As with the Dorim 1 Watershed, the parameters of the SWMM model were divided into topographic and hydrologic parameters. For the topographic parameters, 1/5000 scale topographic map and GIS were used. For the hydrologic parameters, the correction range from the lower limit value to the upper limit value of the parameter was used because of the broad range according to the user's subjective judgment and the same material.

**Table 3.** Estimation results of the main parameters (Gunja watershed).

| Main Input Parameters of XP-SWMM | | Initial Parameters Value | Estimation Parameters Value |
|---|---|---|---|
| Catchment parameters | Width (m) | 25.0~1207.24 | 25.0~1207.24 |
| | Impervious area (%) | 22.4~100.0 | 22.4~100.0 |
| | Slope (m/m) | 0.002~0.108 | 0.002~0.108 |
| Channel parameters | Circular Size (D mm) | D450~D1000 | D200~D1000 |
| | Rectangular Size (m × m) | 1.3 × 1.3~2.5 × 2.5 | 1.3 × 1.3~2.5 × 2.5 |
| | Manning roughness of channel | 0.014~0.020 | 0.020 |
| Infiltration parameters | Decay rate (Horton's) (1/s) | 0.001 | 0.00056 |
| | Ultimate infiltration rate (Horton's) (mm/h) | 10.0 | 12.7 |
| | Manning roughness of impervious area | 0.014 | 0.020 |
| | Depression storage of impervious area (mm) | 0.0 | 1.3 |

### 3.2.3. Gildong Watershed

The Gildong Watershed, located in Gangdong-gu, Seoul, has a total watershed area of 8.6948 km$^2$ and is composed of four drainage areas: Gildong, Myeongil, Gildonggoji, and Seongnae. The highland pipeline is directly discharged to the Seongnaecheon, and the others are discharged to the Seongnaecheon through the Seongnae rainwater pump station. The Gildong Watershed also has a history of an area of 0.1893 km$^2$ being flooded by the rainfall event on 20 and 21 September 2010.

The input data of the XP-SWMM for 1389 nodes and 1246 links were composed using the pipe network data of Seoul Metropolitan City. For 2D inundation analysis, topographic data were constructed using a numerical map of 1:5000 scale. The grids were composed of about 27,000 cells of 15 × 15 m size each due to the large size of the Gildong Watershed (Figure 13). Because there are no discharge data at the time of flood damage in the Gildong watershed, the input parameters were estimated by considering the topographic data and the operating conditions of the Seongnae rainwater pump station that is operating in the watershed (Table 4).

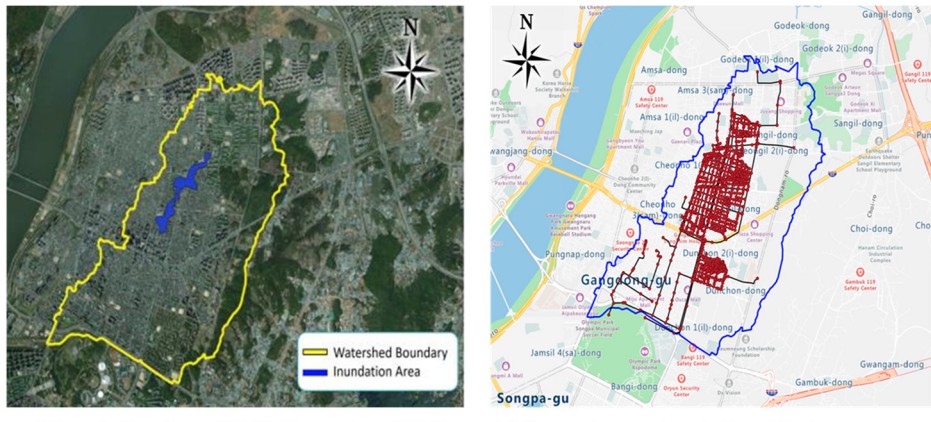

(a) Inundation Area (20-21 September 2010)  (b) Sewer Pipe network Input Data

**Figure 13.** GIS map of the Gildong watershed.

**Table 4.** Specifications of the Seongnae rainwater pump station.

| Drainage System | Watershed Area (km²) | Benefiter Area (km²) | Pump (HP × EA) | Discharge (m³/min) | Storage Capacity (m³) | Reservoir Water Level (m) | | |
|---|---|---|---|---|---|---|---|---|
| | | | | | | H.W.L | L.W.L | Depth |
| Seongnaecheon | 6.34 | 1.13 | 950 × 10 1110 × 3 | 4500 | 237,000 | 13.0 | 9.5 | 3.5 |

Table 5 shows the estimation results of the major input parameters of the Gildong watershed using topographic data, past inundation trace maps, and pump station operating conditions. For the topographic parameters, the 1/5000 scale topographic and GIS maps were used. For hydrological parameters, the correction range from the lower to the upper limit values of the parameter was used due to the broad range according to the user's subjective judgment and the same material.

**Table 5.** Estimation results of the main parameters (Gildong watershed).

| Main Input Parameters of XP-SWMM | | Initial Parameters Value | Estimation Parameters Value |
|---|---|---|---|
| Catchment parameters | Width (m) | 4.0~126.0 | 4.0~126.0 |
| | Impervious area (%) | 37.8~98.0 | 37.8~98.0 |
| | Slope (m/m) | 0.005~0.200 | 0.005~0.200 |
| Channel parameters | Circular Size (D mm) | D300~D1600 | D300~D1600 |
| | Rectangular Size (m × m) | 0.8 × 1.3~3.5 × 2.5 | 0.8 × 1.3~3.5 × 2.5 |
| | Manning Roughness of channel | 0.015~0.020 | 0.020 |
| Infiltration parameters | Decay rate (Horton's) (1/s) | 0.001 | 0.015 |
| | Ultimate infiltration rate (Horton's) (mm/h) | 10.0 | 4.5 |
| | Manning roughness of impervious area | 0.014 | 0.014 |
| | Depression storage of impervious area (mm) | 0.0 | 3.3 |

### 3.3. Analysis of Flood Area According to Whether or Not the Head Loss Coefficient Is Applied

As previously discussed, when the head loss coefficient is applied to surcharge manholes in urban flood areas analysis, it affects the surcharge manhole and the surrounding manholes as well. This causes additional surcharge, and the flood area resulting from the overflow of some surcharge manholes requires larger simulation. In the XP-SWMM, each manhole is processed as a node, and the energy loss in the manhole cannot be considered. Thus, a method for replacing the head loss coefficient of manhole by applying the head loss coefficient to the inlet and outlet losses of surcharge sewers was selected to simulate the change of flood area due to the application of the head loss coefficient. For this, the surcharge sewer was estimated by checking the dynamic cross-section of the link in the primary simulation results of the XP-SWMM model. Then, the energy head loss coefficients of the inlet/outlet losses were input by dynamic flow analysis of the estimated surcharge sewer. In the case of the surcharge circular manhole, 0.31 and 0.30 out of the mean head loss coefficient 0.61 were applied to inlet and outlet losses, respectively. In the case of surcharge square manhole, 0.34 each out of the mean head loss coefficient 0.68 was applied to the inlet and outlet losses, respectively. Furthermore, to compare the flood area according to the change in the head loss coefficient, the flood areas obtained by applying the mean head loss coefficients 0.2, 0.4, 0.6, and 0.8 were compared to the flood areas obtained by applying the head loss coefficient determined by hydraulic experiments.

For this, discharge and inundation analyses were performed using the observed rainfall of the torrential rains that caused flood damages to the target watersheds on 20 and 21 September 2010 in Figure 8. The rainfall event in Figure 7 had a return period of 100 to 200 years, with a total rainfall of 259 mm for a duration of 30 h. This rainfall left serious flood damages in Seoul and the capital area, including the Gwanghwamun Intersection, Hwagok-dong in Gangseo-gu, Sinweol-dong in Yangcheon-gu, and Sadang Intersection in Gwanak-gu.

### 3.3.1. Dorim 1 Watershed

The flood area simulated using XP-SWMM and the observed flood area of the Dorim 1 watershed were compared using the inundation trace map provided by the public data portal of the Ministry of the Interior and Safety (Figure 9a).

Figure 14 compares the flood area simulated using XP-SWMM with the observed flood area for the rainfall event on 20 and 21 September 2010 according to the application of the head loss coefficient to the Dorim 1 watershed, for which circular manholes and square manholes are mixed. The mean head loss coefficient 0.61 was applied to circular manhole, and 0.68 was applied to the square manhole. The results obtained by applying the mean head loss coefficients of 0.2 to 0.8 were also compared.

Table 6 shows the comparison of the inundation area according to the application of the head loss coefficients of the Dorim 1 watershed. The inundation analysis result showed that when the head loss coefficient was not applied, the simulated flood area was 0.2086 km$^2$, which was underestimated when compared to the observed flood area of 0.3565 km$^2$. The rate of concordance between the observed and simulated flood areas was approximately 58.51%. Comparison results when the mean head loss coefficient was varied between 0.2 and 0.8 show that the size of the flood area increases with the mean head loss coefficient. Furthermore, when the calculated mean head loss coefficients of 0.61 and 0.68 were applied for hydraulic experiments, the flood area was 0.3506 km$^2$. Thus, it showed a rate of concordance of 98.35% with the past flood area of 0.3565 km$^2$. When compared to the past inundation trace map, the past flood areas for the Dorim 1 watershed were indicated separately, showing some differences with the simulated results. However, the flood areas were relatively similar because the same floods occurred in the main flooding sections.

**Table 6.** Change of inundation area by the application of head loss coefficients (Dorim 1).

| Head Loss Coefficients Result | None | 0.20 | 0.40 | 0.60 | 0.61, 0.68 | 0.80 |
|---|---|---|---|---|---|---|
| Number of Flooded Cells | 1213 | 1526 | 1819 | 2005 | 2039 | 2172 |
| Simulated Area (km$^2$) | 0.2086 | 0.2625 | 0.3128 | 0.3448 | 0.3506 | 0.3735 |
| Observed Area(km$^2$) | | | | 0.3565 | | |
| Rate of Concordance (%) | 58.51 | 73.63 | 87.74 | 96.72 | 98.35 | 104.79 |

### 3.3.2. Gunja Watershed

In this section, the changes of flood area according to the application of the head loss coefficient for the Gunja watershed were analyzed. The input data were simplified around the main storm-water sewers compared to the input data of the Dorim 1 watershed. Thus, a rational discharge analysis method in a simplified pipe network could be suggested by using the result of analyzing the flood area changes of the Gunja watershed.

Figure 15 shows the flood area simulated using XP-SWMM for the rainfall event on 20 and 21 September 2010 according to the application of the head loss coefficient in the Gunja watershed. Because only circular manholes are installed in the Gunja watershed, the calculated mean head loss coefficient of 0.61 was applied (as determined from hydraulic experiments), and the mean head loss coefficients of 0.2 to 0.8 were applied for comparison.

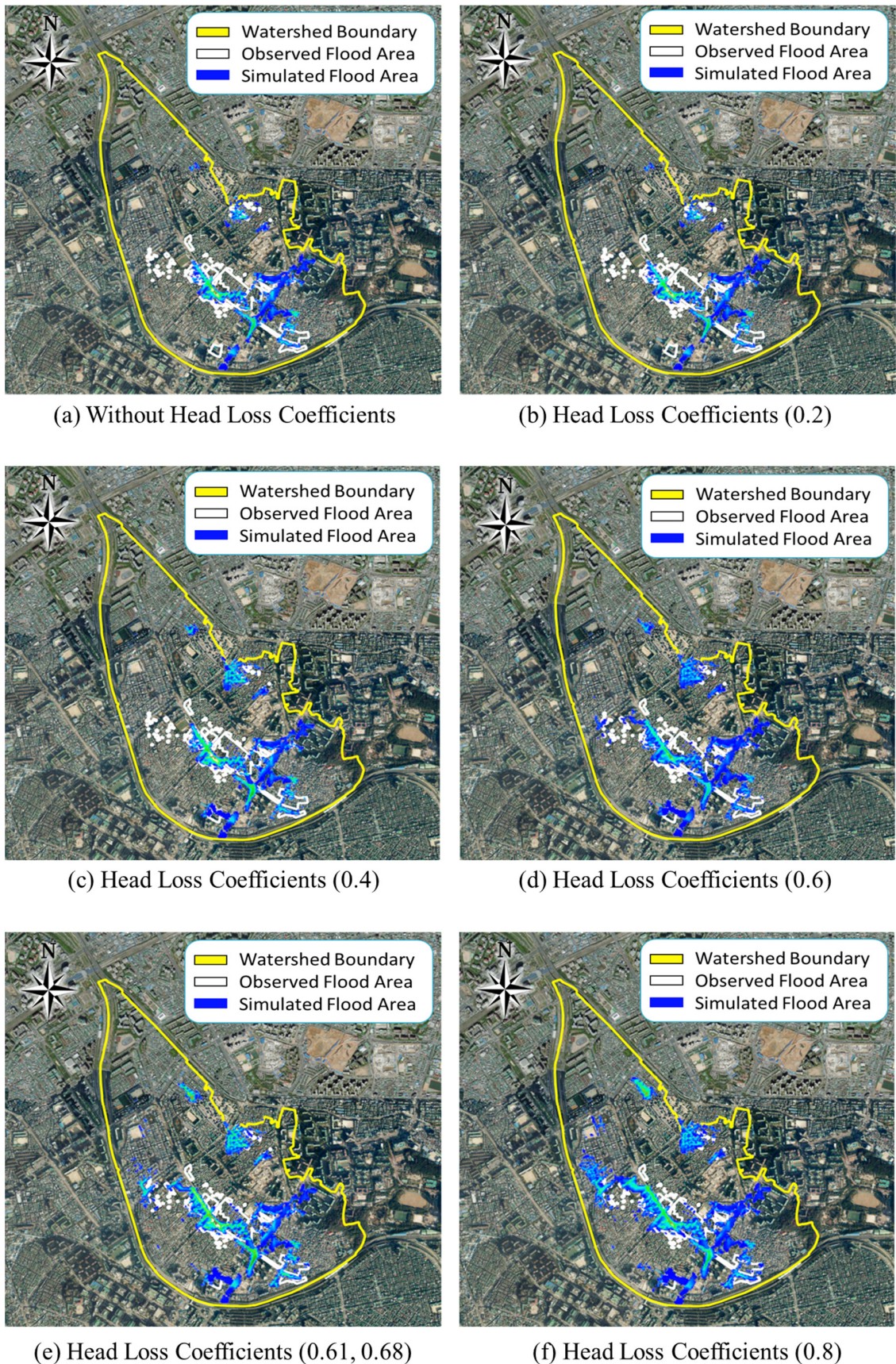

**Figure 14.** Comparison of the observed and simulated inundation area (Dorim 1 watershed).

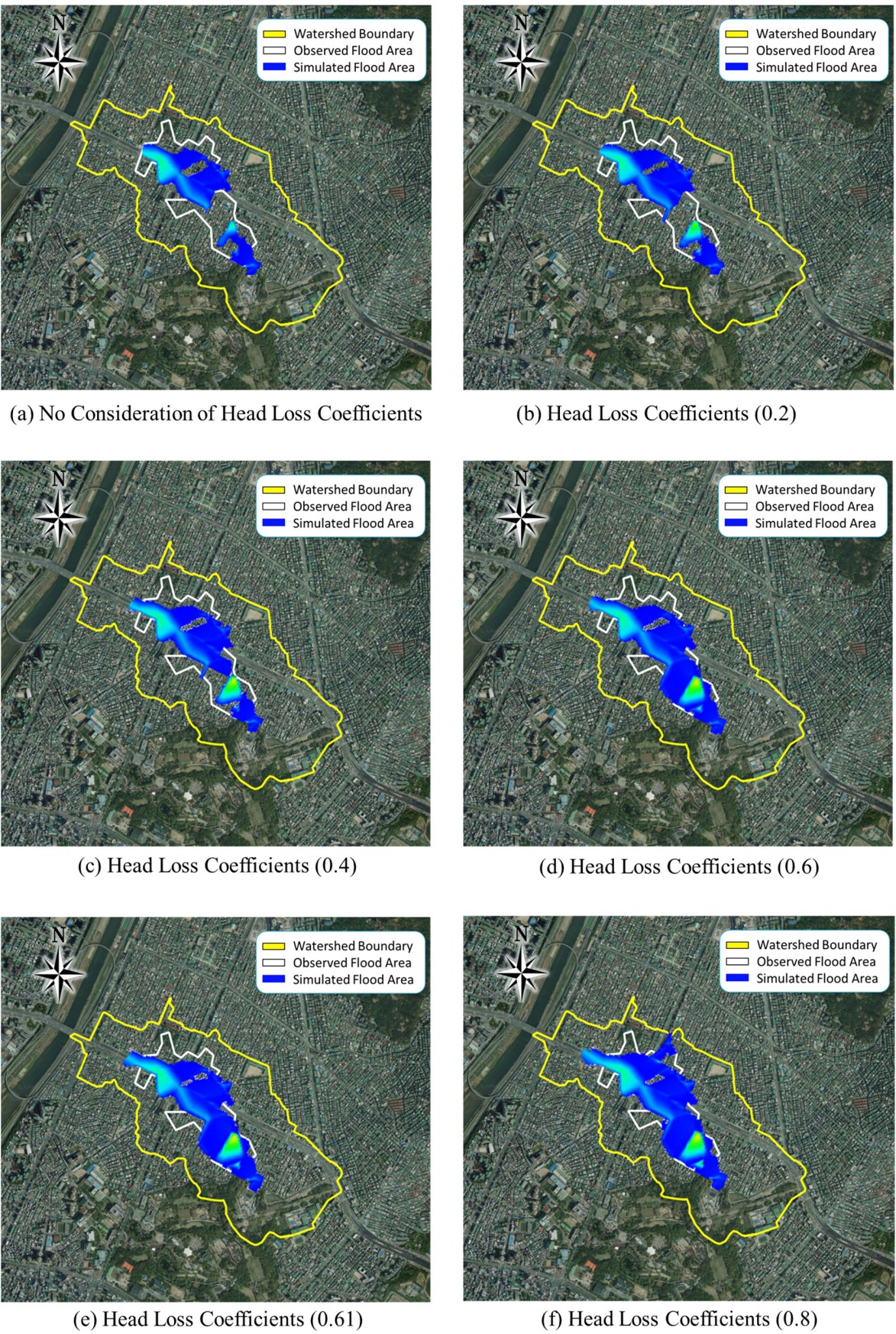

**Figure 15.** Comparison of the observed and simulated inundation area (Gunja watershed).

Table 7 shows the comparison of the inundation area according to the application of the head loss coefficients of the Gunja watershed. The result of the inundation analysis shows that, when the head loss coefficient was not applied, the shape of the flood area around the Gunja Station appears similar, whereas the simulated flood area was 0.1438 km$^2$. Thus, the flood area was underestimated when compared to the observed one of 0.1917 km$^2$, with only 75.01% rate of concordance. Similar to the result of the Dorim 1 watershed, the discharge and flood area became larger as the mean head loss coefficient increased. When the mean head loss coefficient 0.61 was applied, the flood area was 0.1852 km$^2$, and the rate of concordance with the observed flood area of 0.1917 km$^2$ was found at 96.61%. A simplified pipe network was applied for the Gunja Watershed. Although the discharge tends to be underestimated when discharge analysis is performed with a simplified pipe network, here, it was increased by the application of the head loss coefficient in discharge analysis. Therefore, it is believed that rational results of discharge analysis could be derived by applying a head loss coefficient to a simplified pipe network.

**Table 7.** Change of the inundation area by the application of head loss coefficients (Gunja watershed).

| Head Loss Coefficients / Result | None | 0.20 | 0.40 | 0.60 | 0.61, 0.68 | 0.80 |
|---|---|---|---|---|---|---|
| Number of Flooded Cells | 1270 | 1348 | 1440 | 1599 | 1636 | 1809 |
| Simulated Area (km$^2$) | 0.1438 | 0.1527 | 0.1629 | 0.1811 | 0.1852 | 0.2048 |
| Observed Area (km$^2$) | | | | 0.1917 | | |
| Rate of Concordance (%) | 75.01 | 79.66 | 85.01 | 94.48 | 96.61 | 106.83 |

### 3.3.3. Gildong Watershed

In this section, the change of flood area according to the application of the head loss coefficient to the Gildong watershed is analyzed. The Gildong watershed has four drainage areas and is characterized by significantly large watershed area compared to the other two target watersheds. Similar to the above analysis, the mean head loss coefficient of 0.61 was applied to the circular manhole, and the mean head loss coefficient of 0.68 was applied to the square manhole. The inundation analysis results were compared by applying the rainfall event on 20 and 21 September 2010, which caused flood damage in the past (Figure 16).

Table 8 shows the comparison of the inundation area according to the application of the head loss coefficients of the Gildong watershed. Following the analysis changes in the flood area of the Gildong watershed according to the application of the head loss coefficient, the larger the head loss coefficient, the larger the flood area became due to the increased discharge. A relatively similar pattern to the past flood area is identified. However, when the head loss coefficient was not applied, the flood area was 0.1154 km$^2$, which was underestimated compared to the past flood area of 0.1893 km$^2$. When the mean head loss coefficient of 0.4 was applied, the flood area was 0.1708 km$^2$, and the rate of concordance was 90.23%. When the mean head loss coefficients of 0.6 or higher were applied, the rate of concordance of the flood area changed to 98–105%. Thus, the flood area did not show significant changes. This is considered the result of almost reaching the limit of the increase for discharge when the head loss coefficients of 0.6 or higher were applied. When the mean head loss coefficients of 0.61 and 0.68 were applied, the rate of concordance of the flood area was overestimated at 106.67%. Furthermore, the area of the Gildong watershed was considerably larger compared to that of the other target watersheds. Additionally, the number of flooded cells was small compared to the flood area, and the area per cell tended to be larger. Thus, it is necessary to set a reasonable watershed size because there will be disadvantages in accuracy compared to other target watersheds, such as excessive increase in flood area.

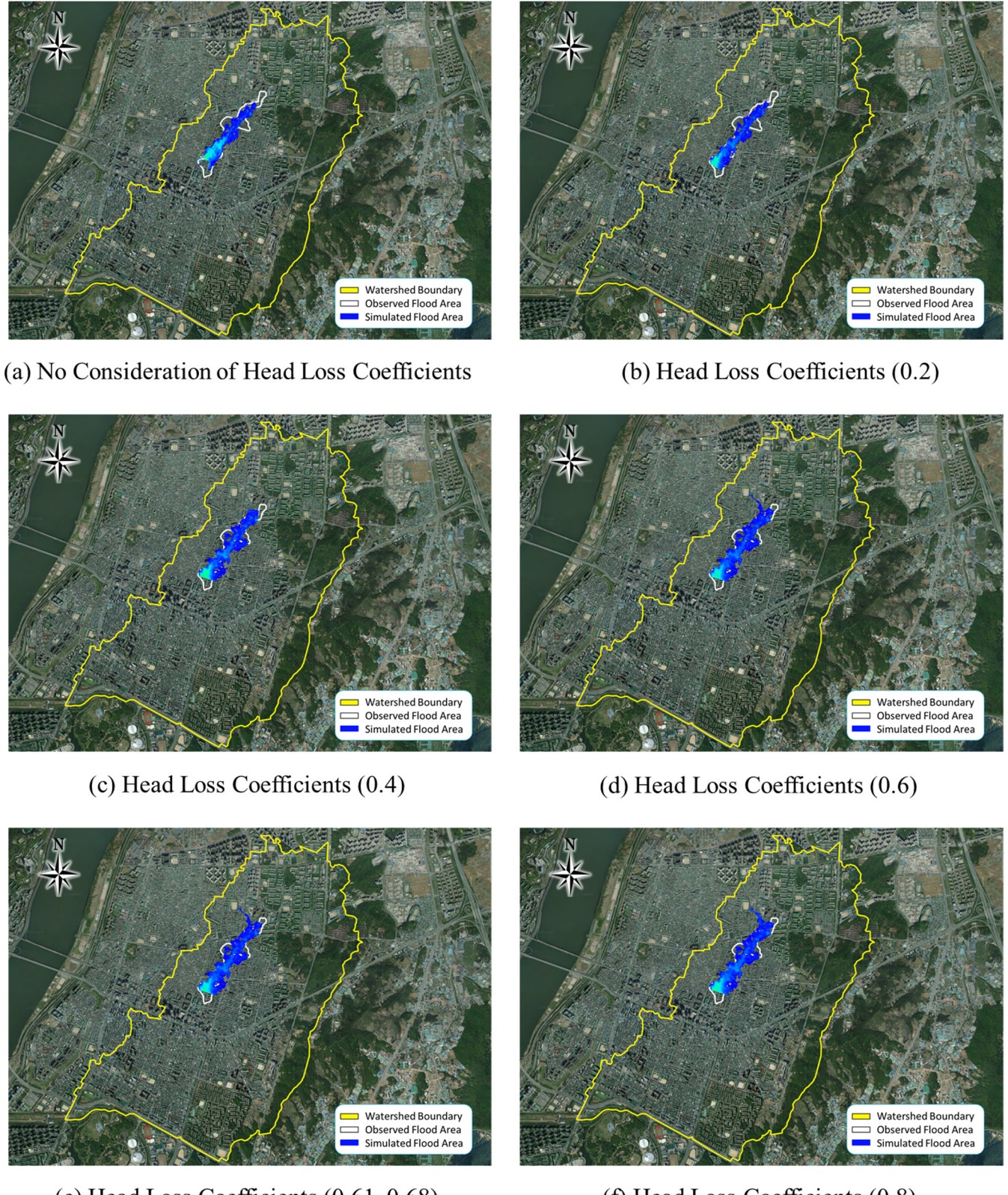

(a) No Consideration of Head Loss Coefficients

(b) Head Loss Coefficients (0.2)

(c) Head Loss Coefficients (0.4)

(d) Head Loss Coefficients (0.6)

(e) Head Loss Coefficients (0.61, 0.68)

(f) Head Loss Coefficients (0.8)

**Figure 16.** Comparison of the observed and simulated inundation area (Gildong watershed).

**Table 8.** Change of the inundation area by the application of head loss coefficients (Gildong).

| Head Loss Coefficients / Result | None | 0.20 | 0.40 | 0.60 | 0.61, 0.68 | 0.80 |
|---|---|---|---|---|---|---|
| Number of Flooded Cells | 513 | 630 | 759 | 828 | 835 | 892 |
| Simulated Area (km$^2$) | 0.1154 | 0.1417 | 0.1708 | 0.1863 | 0.1879 | 0.2005 |
| Observed Area (km$^2$) | | | | 0.1893 | | |
| Rate of Concordance (%) | 60.96 | 74.85 | 90.23 | 98.43 | 99.27 | 105.92 |

## 4. Accuracy of Inundation Result Using LSSI

The Lee–Salle shape index (LSSI) method measures the accuracy of the spatial location by calculating the area of intersection between the comparison targets. In this study, the accuracy of the spatial location between two data was measured by using the result of inundation analysis determined by applying the head loss coefficient of the surcharge manhole and the inundation trace map. The LSSI method is used to measure the location accuracy of the reference data and compare it to other data using values calculated in the form of an index between 0 and 1. The closer the value is to 1, the higher the spatial location accuracy is [33]. The equation for LSSI is given by:

$$LSSI = \frac{A \cap B}{A \cup B},\tag{4}$$

where *A* denotes the inundation trace map, and *B* is the range of inundation analysis result determined by applying the head loss coefficient of manhole.

Due to the above characteristic, the LSSI method is known to be highly efficient for referring to spatial correspondence [34,35].

Table 9 shows the accuracy according to the LSSI range.

**Table 9.** Suggested evaluation criteria of the LSSI method.

| Range of LSSI | Degrees of Accuracy |
|---|---|
| 40.0 over | Excellent |
| 30.0 over | Good |
| 20.0 over | Fair |
| 10.0 over | Poor |
| 5.0 over | Fail |

The LSSI analysis was performed using the result of the XP-SWMM inundation analysis determined by applying the inundation trace map and surcharge straight path manhole head loss coefficient of the Dorim 1, Gunja, and Gildong Watersheds (Figure 17).

**Table 10.** Result of LSSI method.

| Watershed | Flood Map (A) | XP-SWMM (B) | A ∩ B | A ∪ B | LSSI (%) | Degree |
|---|---|---|---|---|---|---|
| (a) Dorim 1 | 0.3565 | 0.3506 | 0.1951 | 0.5865 | 32.27 | Good |
| (b) Gunja | 0.1917 | 0.1852 | 0.1277 | 0.2398 | 53.25 | Excellent |
| (c) Gildong | 0.1893 | 0.1879 | 0.1348 | 0.2919 | 46.19 | Excellent |

## (a) Dorim 1 Watershed

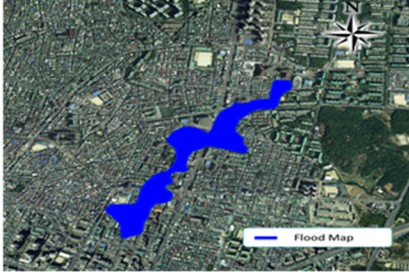

Flood Map       XP-SWMM       Comparison of flood area

## (b) Gunja Watershed

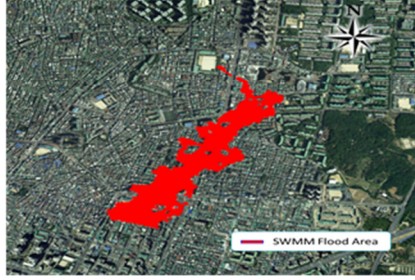

Flood Map       XP-SWMM       Comparison of flood area

## (c) Gildong Watershed

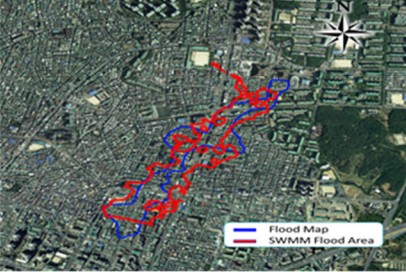

Flood Map       XP-SWMM       Comparison of flood area

**Figure 17.** Comparison of inundation area.

The LSSI analysis for the three watersheds indicated that the Dorim 1 watershed was analyzed as Good with 32.27%, the Gunja watershed as Excellent with 53.25%, and the Gildong watershed as excellent with 46.19% (Table 10). Thus, all the three watersheds were analyzed as good or higher grades in the LSSI analysis result. Consequently, the inundation pattern of the inundation trace map was found to be similar to the inundation analysis of the XP-SWMM model determined by applying the surcharge straight path manhole head loss coefficient.

## 5. Conclusions

In general, the discharge tends to be underestimated when urban inundation analysis is conducted with a simplified pipe network. However, in this study, it had results that the discharge increased when the head loss coefficient was applied to discharge analysis. Therefore, when the flood area is simulated by using a simplified pipe network, a rational

result of discharge analysis could be derived by applying the head loss coefficient of the surcharge manhole.

As a result of conducting hydraulic experiments, the mean head loss coefficients of surcharge straight path (circular and square) manholes were determined as 0.61 and 0.68, using the relationship between observed head loss and velocity head. Performing inundation analysis by applying the head loss coefficient (0.61 and 0.68, which were determined by hydraulic experiments) for three watersheds, the flood area predicted by the XP-SWMM model without considering the head loss coefficient in surcharge manhole showed a significant difference from the observed flood area. However, when it was predicted by applying the head loss coefficient, a result close to the observed flood area could be obtained.

This result suggests that the construction of accurate input data and the application of head loss coefficient can improve the accuracy of discharge analyses.

Therefore, it is believed that more realistic flood area analysis is possible by applying the head loss coefficient when performing flood area analysis of urban watersheds using XP-SWMM. However, the results may appear differently depending on various regional factors and the level of pipe network simplification. Furthermore, the inundation depth could not be reviewed because the inundation depth data could not be obtained from the past inundation trace maps. Therefore, additional research considering these conditions is necessary.

**Author Contributions:** Conceptualization, C.K. methodology, J.K.; software, S.L.; validation, S.L.; formal analysis, C.K.; investigation, I.Y.; resources, I.Y.; data curation, C.K.; writing—original draft preparation, C.K.; writing—review and editing, J.K.; visualization, S.L.; supervision, J.K.; project administration, J.K.; funding acquisition, I.Y. All authors have read and agreed to the published version of the manuscript.

**Funding:** This research was funded by the Korea Agency for Infrastructure Technology Advancement (KAIA) grant funded by the Ministry of Land, Infrastructure and Transport, grant number 22CTAP-C164140-02.

**Institutional Review Board Statement:** Not applicable.

**Informed Consent Statement:** Not applicable.

**Data Availability Statement:** No new data were created or analyzed in this study. Data sharing is not applicable to this article.

**Conflicts of Interest:** The authors declare no conflict of interest. The funder had no role in the design of the study; in the collection, analyses, or interpretation of data; in the writing of the manuscript; or in the decision to publish the results.

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
