# Peer review of "Application of Head Loss Coefficient for Surcharge Straight Path Manhole to Improve the Accuracy of Urban Inundation Analysis"

_water, doi:10.3390/w14172725_

Round 1
Reviewer 1 Report
The studies were carried out on a topical topic and are of scientific and practical interest.
There are a number of comments on the work.
191. The authors used the Froude test in their modeling. It is necessary to indicate the specific values ​​of the criteria for the model and natural structure in order to justify the modeling scale of 1/5.
The accuracy of the hydraulic experiment is not specified. What measurement errors are taken into account by the second? There are errors in direct and indirect measurements in a hydraulic experiment. To indicate the experimental average value of the coefficient of hydraulic resistance without the boundaries of the confidence interval, in our opinion, is not entirely correct. It is necessary to analyze the errors and give the value of the drag coefficient with the boundary of the confidence interval for a confidence level of 95%.
For the regression lines shown in the figures, it is necessary to indicate the regression equations and provide statistical estimates of the quality of the regression lines. For 8 and 9 points, on which the equations are built, this is extremely important.
Reviewer 2 Report
revisions.
Hereinafter, the major recommendation for the authors in the text is only a sample of the issues.
o Rewrite the first part of the introduction. From line 26 to 71 I only see one cited author. You should support your introduction with the appropriate citations.
o In Line 158 to 161, mention the appropriate units of delta h, k and V.
o Enhance the quality of Figure 2 and Figure 3.
o Citation needed for Lines 183 to 185.
o The legend for Figure 4a and 4b are not clear, enhance the quality of this figure too.
o Citation required for the equations 248 to 257.
o The legend of Figure 12 a to 12 f are not readable at all.
o Figure 13, 14 the same as before.
o Conclusion is not a summary of your article, rewrite and abbreviate the conclusion focusing only on your importance of finding results.
Best Regards
Round 2
Reviewer 1 Report
Thanks for the fixes and additions
Reviewer 2 Report
The author(s) edit all the previously required comments, and I accept the manuscript for publication.